# High-Intensity Interval Training During Cancer Prehabilitation May Improve Cardiorespiratory Fitness: A Meta-Analysis

**DOI:** 10.3390/healthcare13233030

**Published:** 2025-11-24

**Authors:** Simone Cuomo, Paolo Riccardo Brustio, Anna Mulasso, Luca Beratto, Christina Dieli-Conwright, Alberto Rainoldi

**Affiliations:** 1Department of Medical Sciences, University of Turin, 10126 Turin, Italy; simone.cuomo@unito.it (S.C.); luca.beratto@unito.it (L.B.); alberto.rainoldi@unito.it (A.R.); 2Neuro Muscular Function|Research Group, University of Turin, 10126 Turin, Italy; paoloriccardo.brustio@unito.it; 3Department of Clinical and Biological Sciences, University of Turin, 10124 Turin, Italy; 4Division of Population Sciences, Department of Medical Oncology, Dana-Farber Cancer Institute, Boston, MA 02215, USA; christinam_dieli-conwright@dfci.harvard.edu; 5Harvard Medical School, Boston, MA 02115, USA; 6Harvard T.H. Chan School of Public Health, Boston, MA 02115, USA

**Keywords:** VO_2_peak, cancer, exercise training, HIIT, prehabilitation

## Abstract

**Highlights:**

**What are the main findings?**
HIIT-based prehabilitation significantly improves VO_2_peak in patients with cancer compared to usual care;Interventions are feasible and safe, with high adherence and low drop-out rates.

**What are the implication of the main findings?**
Incorporating HIIT into prehabilitation can efficiently optimize fitness within limited pre-treatment timeframes;HIIT may enhance postoperative outcomes and quality of life while supporting long-term healthcare sustainability; however, standardized protocols and further research are needed to optimize exercise prescriptions across cancer types.

**Abstract:**

**Background/Objectives**: Cardiovascular disease is the leading cause of non-cancer mortality in cancer survivors. Exercise interventions are widely used to enhance cardiorespiratory fitness, typically assessed by VO_2_peak, which predicts postoperative complications and poorer clinical outcomes. Prehabilitation provides an opportunity to optimize health. Given time constraints, high-intensity interval training (HIIT) may represent a time-efficient strategy to improve fitness during prehabilitation. This meta-analysis examines the effects of HIIT-based prehabilitation versus usual care on VO_2_peak in cancer patients. **Methods**: A systematic search was conducted in Cinahl, Embase, PubMed, Scopus, and Web of Science from database inception to August 1, 2024 using terms related to cancer, prehabilitation, and HIIT. Random-effects meta-analysis was performed on studies assessing the effects of HIIT versus usual care on VO_2_peak in adults with cancer undergoing prehabilitation. Seven studies comprising 352 participants (aged 56–73 years) with mixed cancer types were analyzed. Methodological quality was assessed using the Cochrane Risk of Bias tool (v2) and the Consensus on Exercise Reporting Template (CERT). The primary outcome was VO_2_peak, analyzed using standardized mean differences (SMD) with 95% confidence intervals (CI). **Results**: The meta-analysis demonstrated a small but statistically significant effect in favor of HIIT over UC (SMD = 0.31, 95% CI = 0.09–0.52, *p* < 0.01), with low between-study heterogeneity (I^2^ = 10%). **Conclusions**: This meta-analysis shows that HIIT-based prehabilitation can improve cardiorespiratory fitness in cancer patients and may provide a clinically relevant, time-efficient strategy to optimize functional capacity before treatment. However, the included studies exhibited substantial clinical heterogeneity, and although all interventions were labeled as HIIT, exercise intensity was not assessed consistently across studies, underscoring the need for cancer-specific randomized controlled trials with standardized HIIT protocols and objective intensity verification.

## 1. Introduction

People diagnosed with cancer often experience a range of adverse outcomes directly associated with the disease or its treatment, affecting both physical (e.g., fatigue, decreased physical functioning, cardiotoxicity) and psychological well-being [1,2].

Cardiovascular disease (CVD) is the leading cause of non-cancer death among cancer survivors [3], who are at higher risk of CVD and related mortality than the general population [4] due to shared risk factors and treatment-induced cardiotoxicity [5].

Cardiorespiratory fitness is inversely associated with cardiovascular disease mortality [6], and VO_2_peak is widely recognized as a key indicator of cardiorespiratory capacity [7] and increasingly used as a clinical biomarker [8]. In patients with cancer, lower preoperative VO_2_peak predicts a higher risk of postoperative complications [9,10], including therapy-related late effects, surgical complications, and survival rates [11,12,13].

The period between a cancer diagnosis and the start of treatment (e.g., surgery, chemotherapy, radiotherapy) is known as prehabilitation. It represents a critical window to enhance overall health and reduce treatment-related impairments through targeted interventions such as exercise [14,15]. Clinical guidelines generally limit the time from diagnosis to treatment to about 31 days [14,16]. While shorter waiting times are beneficial for prognosis, they also reduce the available period to induce the physiological adaptations targeted by exercise interventions, thereby limiting the potential magnitude of improvement in physical conditioning. This condensed timeframe for supportive care interventions makes it challenging to achieve the desired physiological adaptations within the prehabilitation phase.

To accelerate physiological adaptations, high-intensity interval training (HIIT), which involves alternating periods of intense work and recovery that vary in intensity and duration [17], has proven feasible and safe [13,18,19] and can efficiently improve physical fitness and health-related outcomes in patients with cancer [18]. Although the correct classification of HIIT requires objective verification of intensity (e.g., heart rate or power output), the existing literature shows marked variability in this regard, which represents an important methodological limitation in this field. Because the prehabilitation phase is constrained by a short time window before treatment, time-efficient exercise models such as HIIT may offer a unique opportunity to maximize physiological gains within clinically relevant limits. However, despite growing evidence supporting exercise prehabilitation, the optimal training modality remains unclear, and the specific benefits of HIIT compared to usual care (UC) during the limited pre-treatment window have not been systematically quantified. Therefore, this meta-analysis aimed to synthesize current evidence on the effects of HIIT-based prehabilitation versus UC on VO_2_peak in adults with cancer. We hypothesized that HIIT would lead to greater improvements in cardiorespiratory fitness than UC, providing a time-efficient strategy to enhance functional capacity before treatment.

## 2. Materials and Methods

### 2.1. Search Strategy

A systematic review was conducted to identify studies published from database inception up to August 1, 2024. The study protocol was prospectively registered in an international register before starting the review (PROSPERO, CRD42023448522, https://www.crd.york.ac.uk/prospero/ (accessed on 14 November 2025)). Five electronic databases, including Cinahl, Embase, PubMed, Scopus, and Web of Science (WOS), were searched using search strategies tailored to each database. The search strategy combined MeSH terms and keywords (synonyms and abbreviations) across three domains: cancer-related terms (“cancer,” “neoplasm,” “tumor,” “malignancy”), prehabilitation/preoperative exercise (“prehabilitation,” “preoperative exercise,” “preoperative rehabilitation”), and high-intensity interval training (“high-intensity interval training,” “HIIT,” “sprint interval training”). Boolean operators (AND, OR) were used to combine terms. References of included articles were also screened to identify additional eligible studies. Full search strings are reported in Appendix A.

### 2.2. Eligibility Criteria

Articles were included in this review if they met the following criteria: (1) Age over 18 years old. (2) Only studies with patients with cancer in a prehabilitative context. (3) Randomized and non-randomized controlled exercise intervention trials with at least one treatment arm being HIIT. (4) Interventions meeting recognized high-intensity thresholds, including ≥85–90% of peak heart rate (HRpeak), ≥80–100% of peak oxygen uptake (VO_2_peak), ≥100% of peak power output (PPO), or a subjective rating of perceived exertion (RPE) between 13 and 15 on the Borg scale, corresponding to high intensity according to ACSM guidelines, as verified from the original studies (see Table 1). When RPE was the only measure reported, its correspondence to ≥85–90% HRpeak was accepted based on ACSM recommendations, provided that the protocol structure (interval duration and recovery) was consistent with standard HIIT formats. (5) Reporting/availability of complete pre- and post-intervention outcomes (VO_2_peak). (6) Used cardiopulmonary exercise testing (CPET) to evaluate VO_2_peak which is widely accepted as the gold standard assessment method for evaluating CRF and assessing the response to rehabilitation programs in various cancer types. (7) Articles written only in the English language. Other sources (e.g., reviews, meta-analyses, abstracts, opinion articles, books, statements, letters, editorials, comments, and non-peer-reviewed journal articles) were excluded.

### 2.3. Article Selection

All potential studies were imported into Covidence (Melbourne, Australia, https://www.covidence.org), and duplicates were removed. A summary of the studies’ screening protocol and selection is provided in Figure 1. The selection process was conducted by two authors (S.C. and P.R.B.) who independently screened the selected title and/or abstract to identify studies that potentially met the inclusion criteria. Then, the same authors examined the full texts of potentially eligible studies. Disagreements were resolved by discussion between the examiners. The corresponding author was contacted if any information was missing from the article. The study was excluded from the analysis if no response or data were available.

### 2.4. Study Quality Assessment

#### 2.4.1. Quality of the Study

The Cochrane Risk of Bias version 2 tool (Cochrane, Oxford, UK. https://www.riskofbias.info/welcome/rob-2-0-tool (accessed on 14 November 2025)) assessed the quality of the reported studies. This tool evaluates five domains of the RCTs to determine their overall risk of bias (RoB). The tool also guides the assessment with a set of signaling questions that require a justified response and straightforward answers and provides an algorithm to summarize the results of each question. Therefore, each domain was assigned three options: (a) Low RoB, (b) Some Concerns, and (c) High RoB. Then, we looked at all the scores and gave a final score for each study based on the algorithm. Finally, the overall rating of each RCT is based on the worst judgment among the five domains. Two independent authors (S.C. and P.R.B.) completed the study quality assessment. The assessors resolved any disagreements by discussion. The RoB assessments were also considered in the interpretive synthesis. Studies with high risk of bias were retained for completeness but given lower interpretive weight when summarizing findings, whereas low- and moderate-risk studies contributed more substantially to the conclusions.

#### 2.4.2. Quality of the Exercise Intervention

This study used the Consensus on Exercise Reporting Template (CERT), applied by author S.C., to comprehensively assess the quality of exercise interventions. The evaluation included 16 standardized questions, each contributing to a total score ranging from 1 to 19. This rigorous assessment provided a robust and systematic analysis and a detailed description of the exercise interventions used in the study (see Appendix A). Mean and range values of total CERT scores were calculated across all included studies to summarize the overall reporting quality. The CERT evaluation also captured whether the HIIT interventions included progression or adjustments in training load over time, allowing for the assessment of intensity adequacy and potential sources of variation in the intervention effects

### 2.5. Meta-Analysis Results

A Microsoft Excel^®^ (Microsoft Corp., Redmond, WA, USA) spreadsheet was created with the following information: study information (i.e., lead author, journal, and year of publication); the aim of the study, population characteristics (i.e., sample size, age, gender, type and stage of cancer if specified); exercise intervention information (i.e., tools used, frequency, intensity, time and type of exercise, adherence if specified); primary outcomes analyzed and measurement tools; results and conclusion. The data were extracted from each section of the manuscript.

After this, a meta-analysis was performed. Specifically, we included only studies comparing HIIT intervention and UC groups in the meta-analysis. Heterogeneity (i.e., the percentage of the total variability in effect size between studies) was evaluated using the I^2^ index. The level of heterogeneity represented by the I^2^ index was interpreted as low (25% to ≤50%), moderate (50% to ≤75%), and large (>75%). For each pooled analysis, the corresponding qualitative interpretation of heterogeneity (i.e., low, moderate, or high) was reported according to the I^2^ value obtained, ensuring a transparent description of between-study variability. A random-effects model was implemented even when substantial heterogeneity was absent (*p* value > 0.10, I^2^ < 50%) due to this procedure providing a more conservative statistical comparison of the difference between intervention and UC groups. In addition to statistical heterogeneity, clinical heterogeneity among studies (e.g., cancer type, exercise frequency and duration, and prehabilitation time window) was qualitatively examined. Due to the limited number of studies, no formal subgroup or meta-regression analyses were performed, but these factors were considered in the interpretation of the finding. Publication bias was assessed through visual inspection of funnel plots and statistically tested using Egger’s regression test, with a *p*-value < 0.05 considered indicative of potential bias. Continuous data were analyzed and reported using standardized mean differences (SMD) and 95% CI. A *p*-value < 0.05 was considered statistically significant. All statistical analyses were conducted using the packages “meta”, “metafor”, and “metacor” of R Project for Statistical Computing (version 4.2.3; R Core Team, Foundation for Statistical Computing, Vienna, Austria).

## 3. Results

### 3.1. Studies Systematically Identified

Figure 1 summarizes the systematic search and study selection process.

The initial database yielded 176 articles from the database search. We excluded 84 articles because they were duplicates. After screening the titles and abstracts of the remaining 92 articles, we excluded 60 articles for the following reasons: 8 were comments, 7 were posters or abstracts, 24 were reviews, 16 were protocols, 2 were case reports, and 3 were not written in English. We assessed the full texts of the remaining 32 articles for eligibility. Of these, we excluded 25 articles because they did not meet the inclusion criteria (*n* = 10), did not have a prehabilitation intervention (*n* = 4), did not evaluate change in VO_2_peak (*n* = 6), did not have a control group (*n* = 4), or did not have data available (*n* = 1). Thus, we included seven articles in the qualitative synthesis. Figure 1 shows the PRISMA flow diagram.

### 3.2. Study Quality

The overall quality of evidence for each outcome is summarized in Figure 2.

Among the identified sources of bias, the most significant was the lack of participant masking. Four studies [12,16,20,23] were assessed as having a high risk of bias, as indicated in Figure 2. One study [21] was found to have some concerns regarding the risk of bias. Only one study [22] was assessed as having a low risk of bias. Given this distribution, studies with higher risk of bias were retained for completeness but interpreted with caution, whereas those with low or moderate risk contributed more substantially to the overall interpretation of the pooled findings. Four included studies were registered in clinical trial registries [12,16,21,22]. However, it is important to note that the remaining three studies [20,23] have not published any registration or study protocol documentation.

### 3.3. Study Description

Table 1 summarizes the characteristics of identified studies according to the following items: study information (i.e., study name and country); participants information (i.e., sample size, age, gender, and cancer type); exercise information (i.e., duration, frequency, work and recovery intensity, and progression); and adherence.

This work included seven studies published relatively recently, ranging from 2015 [24] to 2023 [21]. The studies were conducted in different regions, with four in the United Kingdom [16,20,22,24] and one each in Switzerland [12], USA [23], and Europe [21].

Among the included studies, all were randomized controlled trials (RCTs), except for study [24], which was designed as a non-randomized, parallel-group, interventional controlled trial.

There was heterogeneity among the studies regarding the types of cancer studied. One study focused on colorectal cancer [24], one study examined liver metastasis from colon cancer [22], while another study focused on allogeneic hematopoietic-cell transplantation [23]. Additionally, three studies addressed urological cancers, with one considering all urological cancers together (i.e., bladder, kidney, and prostate) [16], one focusing on bladder cancer [20], and another concentrating on prostate cancer [21]. One study included patients with lung cancer [12]. Notably, one study proposed prehabilitation after neoadjuvant therapy [24].

Sample sizes across studies ranged from 28 [23] to 151 [12], and the mean sample size across the studies was approximately 55 ± 43. Several studies included older adults, with mean ages exceeding 70 years.

While the overall sample sizes vary across studies, it is important to highlight that gender distribution was unbalanced. In most studies [12,16,20,22,23,24], the proportion of females was lower than that of males. Notably, there was a significant difference among studies where 90% of the HIIT population consisted of females, compared to other studies with a more comparable proportion of female and male participants.

### 3.4. Quality of the Exercise Intervention

#### 3.4.1. HIIT Programs Description

Except for Wood et al. [23], which proposed a home-based HIIT where participants could choose one or more modes of exercise (from a list that included walking, jogging, running, cycling, elliptical, and stair climbing), all the studies utilized supervised HIIT training with a cycle ergometer. Despite this, the interventions exhibited substantial variability between and within studies due to the dynamic nature of the prehabilitation period. Indeed, this period presents a variable window of opportunity for implementing exercise programs (i.e., various factors influence the period between diagnosis and treatment initiation). Consequently, certain variables, such as the duration of the intervention period, could not be controlled. For more information, see the Appendix A.

#### 3.4.2. Training Duration and Frequency

Among the analyzed studies, the intervention duration exhibited variability, ranging from a minimum of 2 weeks, which was proposed as the minimum duration for interventions planned between 2 and 8 weeks [21], to 12 weeks, which was proposed as the maximum duration for interventions scheduled between 5 and 12 weeks [23]. Considering the limited timeframe available for the prehabilitation period, the frequency of the exercise program becomes crucial. The minimum recommended frequency was two times per week [12,20]. However, the majority of the studies (i.e., five studies) suggested a frequency ranging between three [22,24] and four times per week [21] while some studies did not specify whether the frequency should be three or four sessions per week [16,23].

#### 3.4.3. Session Duration

There were significant variations in training session duration across studies. For example, Blackwell et al. [16] suggested the shortest HIIT sessions, which lasted 12.5 min and consisted of five sets of 1 min of high-intensity exercise followed by 1.5 min of recovery. In contrast, Banerjee et al. [20] proposed the longest, approximately 45 min. The protocol involved six interval-training sessions, each lasting 5 min with 2.5 min of rest. The training session duration in Licker et al. [12] was similar to that in Blackwell et al. [16], except that it was repeated twice. In Licker et al. [12], the HIIT protocol consisted of two 10 min sessions, each including 15 s of high-intensity exercise followed by 15 s rest, with a 4 min rest period between sets. Consequently, the total duration of the HIIT session amounted to 24 min. Additionally, the majority of the studies included in this analysis suggested HIIT durations of approximately 25 [12,21] to 30 min [22,23,24] for their respective protocols.

Overall, HIIT protocols varied substantially in total session duration (ranging from 12.5 to 45 min) and in the structure of work and recovery intervals. This heterogeneity in training volume and session design may have influenced the magnitude of the observed effects. Because the number of included trials was limited, formal subgroup or sensitivity analyses were not feasible. Nonetheless, this protocol variability likely contributed to between-study differences and may have attenuated the pooled effect. Future meta-analyses including a larger number of standardized HIIT interventions will be able to investigate this through adequately powered subgroup or sensitivity analyses.

#### 3.4.4. Intensity, Recovery, and Progression

Intensity monitoring is not standardized across different studies and was evaluated differently. Most of the studies [12,16,21,24] utilized a percentage of maximum work rate (WR) to determine the intensity of the training. Nevertheless, among the studies analyzed, three did not report the progression of the intervention [20,22,23], while four studies provided detailed information on the progression [12,16,21,24]. Specifically, Blackwell et al. [16] proposed an intervention consisting of five 1 min intervals at 100–110% maximal WR, separated by 1.50 min of active unloaded recovery. Licker et al. [12] included training protocols with two 5 min intervals of 10 × 15 s at 80–100% WR peak with 15 s recovery. Recovery between the two intervals was 4 min. The progression was determined by adjusting the work rate during each session, guided by the physiotherapist. This adjustment aimed to target near-maximal heart rates towards the conclusion of each series of sprints, considering the individual’s exercise response.

In Djurhuus et al. [21], the exercise program consisted of four periods, where period 1 (week 1) was an adaptation period composed of four cycles of HI at 100% WR peak. Periods 2 (week 2) and 3 (week 3 and 4) consisted of 4 and 5 cycles at 110% and 120% WR peak, respectively. Lastly, period 4 (weeks 5 to 8) comprised six cycles at 120% Wpeak. In West et al. [24], the interval-training program consisted of the first two sessions in a total of 20 min of alternating 3 min moderate intensity intervals set at 80% of WR at VO_2_ at lactate threshold to 2 min severe intensity intervals set at 50% of the difference in work rates between VO_2_peak and VO_2_ at lactate thresholds. This increased to 40 min (6 × 3 min intervals at moderate intensity and 6 × 2 min at severe intensity). Banerjee et al. [20] utilized the rating of perceived exertion (RPE) scale (6–20), with the exercise program consisting of six 5 min intervals at 13–15 RPE, and the exercise program was progressed by gradually increasing the load on the flywheel to maintain the target perceived exertion. It should be noted that in one study [20], exercise intensity was monitored using the Borg RPE scale, without a formal familiarization procedure, rather than direct physiological measures. While this subjective approach has been validated and aligns with ACSM recommendations, the lack of objective verification may have allowed some sessions to be performed at slightly lower intensities, potentially contributing to the modest overall effect observed. Dunne et al. [22] proposed 30 min interval training structured with high intensity at >90% VO_2_peak followed by active recovery at <60% VO_2_peak. However, the study lacked details on the number of intervals, the duration of both the intervals and the active recovery periods, and the progression scheme. Finally, one study employed maximum heart rate (MHR). In particular, Wood et al. [23] suggested 30 min interval training with 5 × 2 min at >80% MHR and 3 min at lower intensity.

#### 3.4.5. Adherence and Drop-Out

Six of the seven studies [12,16,20,21,22,24] reported adherence to the exercise intervention. Generally, adherence was expressed as a percentage of the total number of scheduled lessons attended. Licker et al. [12] reported an adherence rate of 87 ± 18%, while West et al. [24] reported 95.5%. In contrast, other articles reported adherence as the percentage of individuals who achieved a minimum number of lessons. For example, Blackwell et al. [16] evaluated adherence by requiring participants to complete at least 10 HIIT sessions, and 84% of the patients met this criterion. In Djurhuus et al. [21], only 55% (11 out of 20 patients) attended at least 75% of the scheduled exercise sessions. Dunne et al. [22] reported that 18 out of 19 patients completed 100% of the sessions. Banerjee et al., [20] indicated that patients completed a median of 8 sessions (range 1–10) and 6–14 sessions, respectively. A notable variability in drop-out rates is evident. The study by Wood et al. [23] stands out with the highest drop-out rate of 53.85% (7 out of 13 participants), suggesting potential challenges in maintaining participant engagement for most of the study. Licker et al. [12] reported a drop-out rate of 8.64% (7 out of 81 participants) in their study, while Banerjee et al. [20] observed a slightly higher rate of 10.34% (3 out of 29 participants). Dunne et al. [22] reported a lower drop-out rate of 5% (1 out of 20 participants), and Blackwell et al. [16] recorded a drop-out rate of 5.26% (1 out of 19 participants). Notably, Djurhuus et al. [21] and West et al. [24] documented no drop-outs, indicating a commendable level of participant engagement and commitment to the intervention across their studies. These variations in drop-out rates highlight the importance of understanding the factors contributing to participant attrition and the need for tailored strategies to improve intervention adherence and minimize drop-outs. Further investigation into the reasons behind these divergent drop-out rates may provide valuable insights for optimizing the design and implementation of future interventions. Data on adverse events are consistent across studies, with very few reports of such events during the intervention phase. 

#### 3.4.6. Meta-Analyses

A total of seven studies were included in the meta-analysis. Meta-analysis indicated that the combined effect size SMD = 0.31 [95% CI 0.09, 0.52], *p* < 0.01, suggesting a statistically significant difference compared to the UC group, and that HIIT was effective in improving VO_2_peak in patients with cancer during prehabilitation period. Although the pooled effect size (SMD = 0.31) reached statistical significance, its magnitude represents a small-to-moderate improvement, which should be interpreted with caution regarding clinical relevance. Nevertheless, even modest improvements in VO_2_peak may contribute to better surgical tolerance and recovery in cancer prehabilitation. The heterogeneity test results revealed that it was low (I^2^ = 10.0%; τ^2^ < 0.04; *p* = 0.36). Forest plot results are presented in Figure 3. Publication bias was assessed through visual inspection of the funnel plot and Egger’s regression test. The plot appeared symmetrical, and Egger’s test showed no significant asymmetry (z = 1.07, *p* = 0.29), indicating no evidence of publication bias (Figure 4). Although visual asymmetry may appear more pronounced when few studies are available (*n* = 7), this pattern is expected in small-sample meta-analyses, and the non-significant Egger’s test supports that publication bias is unlikely.

## 4. Discussion

This meta-analysis aims to summarize existing evidence on the effects of HIIT in patients with cancer during prehabilitation. For this purpose, considering seven publications, we analyzed the difference in VO_2_peak values between HIIT and UC interventions. The main findings of the study were that HIIT interventions effectively improved VO_2_peak during cancer prehabilitation. Again, the high adherence, low drop-out rates, and the absence of adverse events observed in the studies confirm the feasibility and safety of HIIT for this specific population. Nevertheless, considerable heterogeneity among studies in cancer type and HIIT modalities (i.e., frequency, intensity, time) highlights the need for additional studies to better define specific guidelines for cancer prehabilitation. This finding directly addresses the research gap outlined in the Introduction—specifically, the absence of a comprehensive and up-to-date synthesis quantifying the comparative effectiveness of HIIT versus usual care within the limited prehabilitation window. By integrating recent trials and applying standardized criteria for high-intensity exercise, this meta-analysis provides a refined estimate of HIIT efficacy in this context and delineates key areas where further evidence is needed to inform clinical implementation.

The present study revealed a small but significant difference between HIIT and UC intervention programs of 0.31 [0.09, 0.52] (see Figure 2) with a low degree of heterogeneity among the considered studies (I^2^ = 10.0%, τ^2^ < 0.0001, *p* = 0.36). Across cancer types, our results indicated differences inVO_2_peak in favor of the HIIT exercise program during prehabilitation intervention, confirming the benefit of this type of intervention. Interestingly, greater standardized mean differences among studies were observed in patients with lower baseline VO_2_peak values (see Figure 2 for more details). Therefore, these findings are crucial in the context of prehabilitation, considering the known inverse association between VO_2_peak and postoperative complication risk [9,11] and survival rates [12,13]. Additionally, a high VO_2_peak level may positively influence activities of daily living [25] and, consequently, the quality of life in patients with cancer [26,27].

Two prior meta-analyses have examined this topic: one by Palma et al. [13] and another by Smyth et al. [19].

Palma et al. conducted a review assessing HIIT’s feasibility, safety, and impact on cardiorespiratory fitness and patient-reported outcomes during the prehabilitation of patients with cancer. One key distinction between their work and ours lies in the definition of prehabilitation. Palma et al. [13] included studies not only during the typical prehabilitation period but also during concurrent treatments like chemotherapy and radiotherapy. These broader inclusion criteria covered interventions of varying durations. In contrast, Smyth et al. [19] focused exclusively on HIIT before surgical resection and combined studies comparing HIIT with both UC and HIIT with moderate-intensity continuous training.

Our results align with Palma et al. [13], who demonstrated that a HIIT program during cancer prehabilitation significantly improved cardiorespiratory fitness. Conversely, our results contrast with Smyth et al. [19], who did not observe differences between pre- and post-exercise intervention effects on VO_2_peak. These discrepancies likely reflect methodological and conceptual differences across reviews. Specifically, Smyth et al. [19] pooled HIIT with MICT comparators, potentially diluting the specific effects of HIIT, whereas the present analysis focused exclusively on HIIT versus UC. Moreover, both previous meta-analyses used a fixed-effect model, while our study employed a random-effects model to provide a more conservative estimate and account for potential between-study variability. Finally, although both studies contributed valuable insights to this field, Palma et al. [13] and Smyth et al. [19] included data only up to March 2020 and April 2021, respectively. Considering the exponential increase in interest about this topic and the consequent rise in the number of publications, an updated analysis was warranted, allowing the inclusion of studies not covered in previous reviews. In addition, our study applied a stricter operational definition of prehabilitation, included studies up to August 2024, and implemented a broader yet methodologically robust synthesis of the available evidence, thereby enhancing both the scope and the interpretive reliability of the current findings.

Several noteworthy findings emerged from the analysis of HIIT characteristics. The data indicated substantial heterogeneity in HIIT protocols across studies. Such differences may reflect the dynamic nature of the prehabilitation period, which provides a variable window of opportunity for implementing exercise programs that can influence improvements in VO_2_peak over the course of the intervention. In particular, variations in training frequency, intensity prescription, session volume, and progression schemes across studies may have influenced the magnitude of the observed effects. These differences likely contributed to the modest pooled improvement in VO_2_peak and should be considered when interpreting the clinical relevance of the findings. Given the variability in cancer type, training intensity, and intervention characteristics among studies, this modest effect likely reflects true clinical heterogeneity rather than a uniform physiological response. Part of this variability may be attributed to differences in intensity verification methods. In particular, one study relied solely on RPE without prior familiarization, which may have introduced minor uncertainty in perceived exertion accuracy. However, because objective measures confirmed high-intensity workloads in most included trials, this limitation is unlikely to have materially influenced the overall conclusions. Therefore, based on these observations, the optimal HIIT parameters for prehabilitation, including duration, volume, exercise type, and timing, remain unclear. Consequently, despite these valuable insights, the precise influence of protocol-related disparities on VO_2_peak outcomes remains uncertain. This variability also limits the clinical transferability of current findings. Because HIIT protocols differed in frequency, progression schemes, and exercise modality, it remains difficult to identify standardized parameters suitable for clinical implementation. Although HIIT appears effective overall, the lack of protocol uniformity weakens the ability to formulate specific, evidence-based recommendations for oncological prehabilitation. It is important to note, however, that our meta-analysis included patients with various cancer types. It is plausible that the effects of HIIT vary according to specific training parameters (e.g., frequency, intensity, progression, volume, exercise type, and timing). In the literature, CVD is the leading cause of non-cancer death among cancer survivors [28], and higher cardiorespiratory fitness is associated with a reduced risk of CVD mortality [6]. HIIT has been shown to increase cardiorespiratory fitness, as assessed by VO_2_peak, comparable to moderate-intensity continuous training, despite requiring substantially less time and total training volume [29].

The effectiveness of HIIT in improving cardiorespiratory fitness is attributed to two main factors: molecular mechanisms (i.e., mitochondrial oxidative capacity and biogenesis) and adaptations in cardiovascular structure and function [30].

From a physiological perspective, HIIT enhances mitochondrial efficiency and stimulates key cellular pathways involved in energy metabolism, such as PGC-1α and AMPK. These adaptations support improved oxygen utilization and overall cardiorespiratory performance, providing a biological rationale for the observed increase in VO_2_peak among patients undergoing prehabilitation [30,31,32].

However, it should also be noted that several trials in the oncology field, including some included in this review, provided limited information on the precise intensity achieved during HIIT sessions. This insufficient reporting constrains the ability to link observed improvements in VO_2_peak to specific high-intensity-dependent physiological mechanisms. Therefore, mechanistic interpretations should be considered exploratory rather than definitive until future studies systematically verify and report exercise intensity through objective physiological measures. 

Moreover, it is important to emphasize that the proposed physiological mechanisms—and the time-efficiency rationale commonly attributed to HIIT—presuppose that exercise intensity truly reached high thresholds. Because not all studies objectively verified this condition, these mechanistic explanations should be interpreted as theoretical models rather than definitive evidence of high-intensity-dependent adaptations. It should also be considered that insufficient control or verification of exercise intensity in a few included studies may have acted as a factor diluting the overall effect size. This possibility should be regarded as a form of conceptual sensitivity, highlighting that the modest pooled improvement in VO_2_peak could underestimate the true potential of well-controlled HIIT interventions. Nevertheless, the consistent direction of the effects across studies supports the robustness of the observed benefit despite this methodological limitation.

These physiological improvements may have meaningful clinical implications, as even modest increases in VO_2_peak before surgery are associated with reduced postoperative complications, faster recovery, and improved overall prognosis in patients with cancer. This supports the integration of HIIT-based prehabilitation as a time-efficient, effective strategy within oncological care pathways.

The mechanisms regulating cardiovascular structure and function adaptations to various forms of HIIT have not yet been comprehensively studied [30]. However, as little as two weeks of Wingate-based HIIT has been reported to increase cardiorespiratory capacity, as reflected by changes in VO_2_peak [33]. Furthermore, Rakobobowchuk et al. [34] reported that, in young healthy men and women, HIIT improved peripheral arterial compliance and endothelial function in the trained limbs, reaching levels comparable to those achieved after a higher volume of continuous moderate-intensity training. Thus, the aforementioned factors could explain why our results showed that, compared with UC, HIIT improved VO_2_peak. Moreover, the greater effectiveness of HIIT in eliciting these physiological adaptations, even compared with conventional moderate aerobic training [35,36], may represent an important advantage in overcoming the time constraints typical of prehabilitation. Moreover, data suggest that HIIT is feasible and safe for patients with cancer. Indeed, adherence issues in prehabilitation programs represent an additional limitation in this population. HIIT’s shorter sessions may offset the longer duration of traditional aerobic exercise sessions and enhance long-term adherence to prehabilitation programs.

Future investigations should increase sample sizes and tailor interventions more precisely to specific populations (e.g., cancer type or baseline fitness level). Moreover, future studies should move beyond simple comparisons of HIIT and UC to explore a broader range of exercise interventions and determine which approaches are most effective for this population. 

The present systematic review and meta-analysis have several limitations. Our analysis included one non-randomized controlled trial [24], which may have influenced the results. Additionally, the number of studies examining HIIT during cancer prehabilitation was limited and heterogeneous with respect to cancer type. This limitation may have affected the statistical power of our analysis. Another important limitation concerns the verification of exercise intensity across studies. While most trials provided objective measures confirming high-intensity workloads, one study relied exclusively on the Borg RPE scale without prior familiarization. This inconsistency may have introduced minor uncertainty regarding the true intensity achieved and should therefore be considered a major methodological limitation affecting the interpretive certainty of the pooled results. Furthermore, some articles did not report complete data required for inclusion in the meta-analysis. Finally, only articles written in English and indexed in Cinahl, Embase, PubMed, Scopus, or Web of Science were included.

## 5. Conclusions

This meta-analysis suggests that incorporating HIIT during prehabilitation in patients with cancer may improve cardiorespiratory fitness, particularly VO_2_peak, supporting its potential feasibility and clinical value as a time-efficient intervention before treatment. However, because the intensity of some included interventions was not objectively verified and considerable heterogeneity exists in exercise protocols and cancer types, these findings should be interpreted with caution. Larger, well-designed randomized controlled trials are needed to confirm these results, clarify the dose–response relationship of HIIT across cancer types, and determine how variations in duration, volume, exercise modality, and timing influence outcomes.

## Figures and Tables

**Figure 1 healthcare-13-03030-f001:**
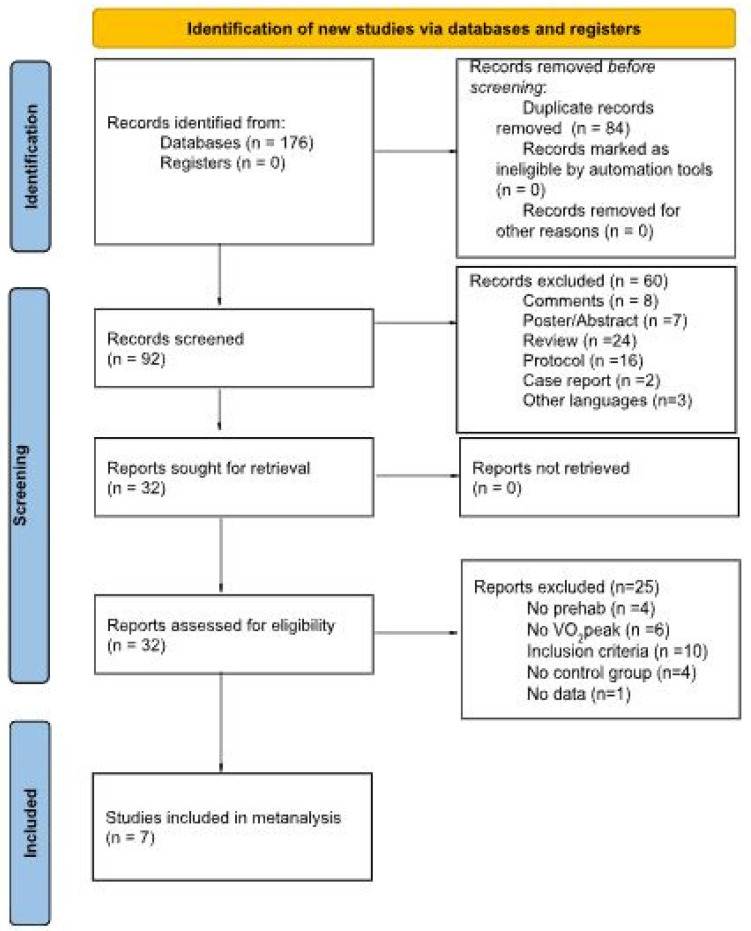
PRISMA 2020 flow diagram illustrating the study selection process for the systematic review and meta-analysis.

**Figure 2 healthcare-13-03030-f002:**
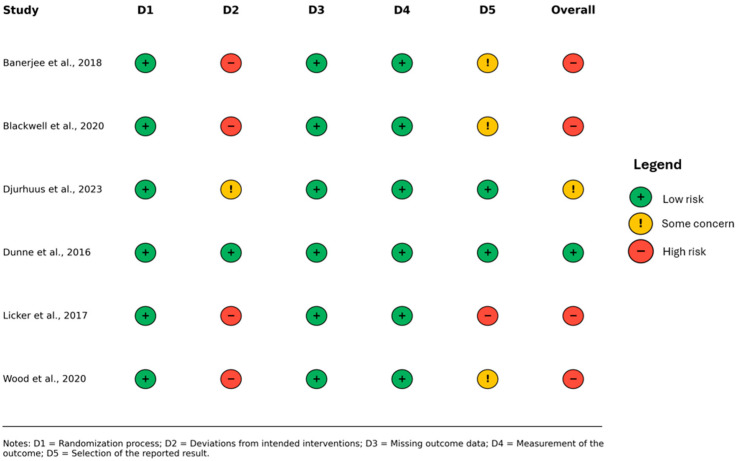
Summary of risk-of-bias judgments for the included studies, assessed using the Cochrane Risk of Bias 2 (RoB 2) tool [12,16,20,21,22,23].

**Figure 3 healthcare-13-03030-f003:**
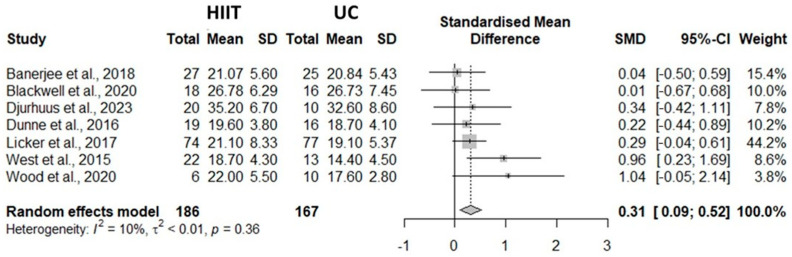
Meta-analysis forest plot results [12,16,20,21,22,23,24].

**Figure 4 healthcare-13-03030-f004:**
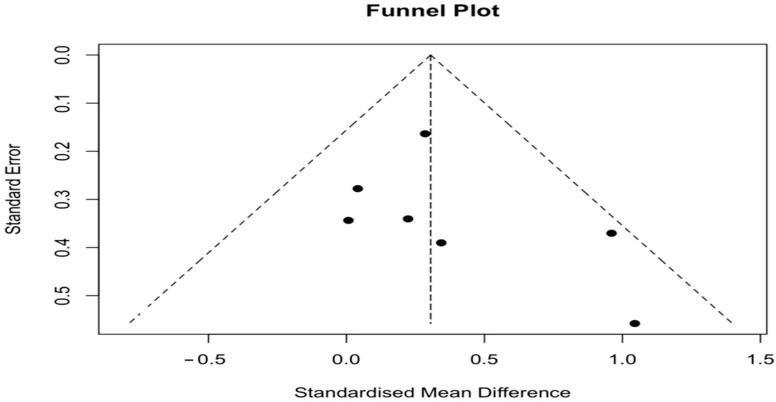
Funnel plot for publication bias in the HIIT vs. UC meta-analysis. No asymmetry was detected (Egger’s test: z = 1.07, *p* = 0.29).

**Table 1 healthcare-13-03030-t001:** Characteristics of the studies included in the meta-analysis.

Study	Participants				Training Characteristics						Adherence
	Cancer Typology	N	Age	Gender (% Female)	Training DURATION (Weeks)	Frequency (n. per Week)	Session Duration	Intensity	Recovery	Progression	
Banerjee et al., 2017 [20]	Bladder Cancer	N = 52 HIIT = 27 UC = 25	HIIT = 72 ± 7 UC = 73 ± 8	HIIT = 10% UC = 13%	3–6	2	~45 min	6 × 5 min at 13–15 RPE Borg	2.5 min light resistance	N.R.	Median of 8 (range: 1–10)
Blackwell et al., 2020 [16]	Urological cancer	N = 34 HIIT = 18 UC = 16	HIIT = 71 ± 2 UC = 72 ± 4	HIIT = 0% UC = 5%	31 days	3–4	12.5 min	5 × 1 min at 100–115% of max load	1.5 min Unloaded active recovery	Increase wattage mid-training to maintain exercise intensity	84% assessed completing ≥ 10 HIIT sessions
Djurhuus et al., 2023 [21]	Prostate cancer	N = 49 HIIT = 20 UC = 29	HIIT = 63 (57–67) UC = 68 (61–70)	HIIT = 0% UC = 0%	2–8	4	20/25 min	4–6 cycles 1 min at 100–120% of Wpeak	3 min at 30% of Wpeak	Wk1: 4 cycles at 100% Wpeak Wk2–4: 4–5 cycles at 110–120% Wk5–8: 6 cycles at 120% Wpeak.	55% (11/20) attended ≥75% exercise sessions during a minimum of 5 Wk
Dunne et al., 2016 [22]	Colon cancer	N = 35 HIIT = 19 UC = 16	HIIT = 61 (56–66) UC = 62 (53–72)	HIIT = 35% UC = 23%	4	10 sessions + 2 recovery sessions	30 min	>90% VO_2_peak	<60% VO_2_peak	N.R.	18/19 completed the 100%
Licker et al., 2017 [12]	Lung cancer	N = 151 HIIT = 74 UC = 77	HIIT = 64 ± 13 UC = 64 ± 10	HIIT = 45% UC = 35%	3–4	2–3	2 × 10 min 4 min rest period between series	15 s at 80–100%WRpeak	15 s	Adjusted during each session to target near MHR	87 ± 18% 8 sessions (IQR 7–10)
West et al., 2015 [24] Non randomized	Colorectal Cancer (after neoadjuvant therapy)	N = 34 HIIT = 22 UC = 12	HIIT = 64 (45–82) UC = 72 (62–84)	HIIT = 36% UC = 31%	6	3	30 min	50% of the difference in work rates between VO_2_peak and VO_2_ at lactate threshold by 2 min intervals	80% of work rate at VO_2_ at lactate threshold by 3 min intervals	Increased from a total of 20 min to 40 min (6 × 3 min intervals at moderate intensity and 6 × 2 min intervals at severe intensity	96%
Wood et al., 2020 [23]	Before allogeneic HCT	N = 28 HIIT = 13 UC = 5	HIIT = 52 (28–73) UC = 72 (62–84)	HIIT = N.R. UC = N.R.	5–12	3–4	30 min	5 × 2 min at ≥80% MHR	3 min at lower-intensity	N.R.	N.R.

Notes: HIIT: high intensity interval training; UC: usual care; N.R.: not reported; WRpeak: work rate peak; HR: heart rate; MHR: maximum heart rate; VO_2_peak: peak oxygen uptake; Wk: week; IQR: interquartile range; min: minutes; N: number of participants; s: seconds; %: percent; HCT: hematopoietic cell transplantation; RPE: rate of perceived exertion.

## Data Availability

No new data were created or analyzed in this study. Data sharing is not applicable to this article.

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
