# Peer review of "High-Intensity Interval Training During Cancer Prehabilitation May Improve Cardiorespiratory Fitness: A Meta-Analysis"

_healthcare, 2025, doi:10.3390/healthcare13233030_

Round 1
Reviewer 1 Report
Comments and Suggestions for Authors
Abstract
The time frame of the literature search is not specified (only “up to August 1, 2024” is mentioned).
The search strategy could briefly summarize the main keywords and MeSH terms used.
The total number of participants, age range, and types of cancer (e.g., breast, prostate, colorectal) should be indicated.
Heterogeneity (I² value) was not reported and should be included.
The statement “More extensive randomized trials are needed” is standard but too general.
A short, practical implication for clinical applications should be added.
Introduction
The introduction is too long and dense.
Lines 71–80: The definition of “prehabilitation” is overexplained and should be shortened.
References [1,2] are cited more than once unnecessarily.
The study’s hypothesis is unclear, as is the specific research gap. These need to be explicitly stated.
Methods
The search strategy includes five major databases (Cinahl, Embase, PubMed, Scopus, WOS).
Registration in PROSPERO (CRD42023448522) strengthens methodological transparency.
Inclusion criteria (age, cancer diagnosis, HIIT, etc.) are appropriate.
CPET as the gold standard assessment is appropriate.
Use of a random-effects model is justified.
Study quality was assessed using the Cochrane RoB 2 tool and exercise intervention quality using CERT — both appropriate.
However, the complete search strings (MeSH + keywords) should be briefly summarized in the main text.
Line 100: A comma is missing between Embase and PubMed.
The time frame of the literature search (starting year) should be indicated.
Although I² threshold levels are mentioned, the degree of heterogeneity (e.g., “moderate heterogeneity was observed”) is not reported and should be clarified.
Publication bias should be evaluated and reported (e.g., funnel plot, Egger’s test).
CERT scoring was conducted, but mean or range values are not reported.
Lines 162–165: The sentence is too long and should be restructured for clarity.
Results
The PRISMA 2020 flow diagram is appropriately used.
However, publication bias analysis is missing and should be included (as previously mentioned, this is important).
Substantial variation among HIIT protocols (e.g., 12-min vs. 45-min sessions) could influence the overall effect.
The impact of this heterogeneity should be analyzed or at least discussed through subgroup or sensitivity analysis.
Although SMD = 0.31 is statistically significant, its clinical significance remains unclear.
The authors should discuss how the observed improvement in VO₂peak might translate into health or functional outcomes.
Discussion
Comparison with Palma et al. and Smyth et al. is useful and theoretically relevant, but the differences between the present study and these prior meta-analyses should be explained more clearly.
This discussion could be linked to the “research gap” mentioned (or added) in the Introduction.
The potential influence of varying HIIT parameters (frequency, intensity, volume, etc.) on outcomes should be discussed (see Lines 380 and 398).
Lines 399–407: The mechanistic explanations involving PGC-1α and AMPK are overly detailed biochemically and distract from the main focus of the paper. These could be summarized or simplified.
The authors should integrate their findings with clinical implications more explicitly, rather than focusing predominantly on molecular pathways.
Limitations and Conclusion
The limitations and conclusions are adequate.
The references are current and relevant.
Author Response
Abstract section
Reviewer 1 – Comment 1:
“The time frame of the literature search is not specified (only ‘up to August 1, 2024’ is mentioned).”
Response:
Thank you for this observation. We agree that the search time frame must be explicitly stated. Therefore, we have revised the ABSTRACT (Methods subsection) to clarify that the search was performed from database inception up to August 1, 2024.
Revised text (ABSTRACT – Methods):
“A systematic search was conducted in Cinahl, Embase, PubMed, Scopus, and Web of Science from database inception up to August 1, 2024.”
Reviewer 1 – Comment 2:
“The search strategy could briefly summarize the main keywords and MeSH terms used.”
Response:
We thank the reviewer for this suggestion. As recommended, we clarified in the Abstract that the search was based on terms related to cancer, prehabilitation, and high-intensity interval training.
Revised text (Abstract – Methods):
...from database inception to August 1, 2024. using terms related to cancer, prehabilitation, and HIIT.
Reviewer 1 – Comment 3
“The total number of participants, age range, and types of cancer (e.g., breast, prostate, colorectal) should be indicated.”
Response:
We thank the reviewer for this comment. A high level of detail regarding each cancer type exceeds the space available in the abstract, where reporting space is limited. However, as suggested, we improved the clarity of the population description by explicitly adding the total sample size and age range in the Abstract. A more detailed breakdown of cancer types is already fully reported in Table 1 in the main manuscript.
Revised text (Abstract – Methods):
“Seven studies comprising 352 participants (aged 56–73 years) with mixed cancer types were analyzed.”
Reviewer 1 – Comment 4
“Heterogeneity (I² value) was not reported and should be included.”
Response:
We thank the reviewer for this suggestion. We have added the I² statistic in the Abstract to more clearly report the degree of between-study heterogeneity.
Revised text (Abstract – Results):
“…with low between-study heterogeneity (I² = 10%).”
Reviewer 1 – Comment 5
“The statement ‘More extensive randomized trials are needed’ is standard but too general.”
Response:
We thank the reviewer for this comment. We agree that this sentence was too generic. We have now revised the conclusion to specify the type of future evidence needed — namely cancer-specific randomized controlled trials with standardized HIIT protocols and objective intensity verification.
Revised text (Conclusions):
“…underscoring the need for cancer-specific randomized controlled trials with standardized HIIT protocols and objective intensity verification to strengthen causal inference and clarify the true impact of HIIT during prehabilitation.”
Reviewer 1 – Comment 6
“A short, practical implication for clinical applications should be added.”
Response:
We thank the reviewer for this helpful suggestion. We agree that a concise practical implication strengthens the clinical relevance of our findings. Accordingly, we have added a brief sentence in the Conclusion highlighting the potential clinical role of HIIT within prehabilitation pathways.
Text inserted (Conclusion section):
."..and may provide a clinically relevant, time-efficient strategy to optimize functional capacity before treatment"
Introduction section
Reviewer 1 – Comments 7, 8, 9 – Introduction (length / definition of prehabilitation / repeated references)
“The introduction is too long and dense.”
“Lines 71–80: The definition of ‘prehabilitation’ is overexplained and should be shortened.”
“References [1,2] are cited more than once unnecessarily.”
Response:
We thank the reviewer for these comments. We agree that the Introduction was unnecessarily long and that certain elements could be streamlined. Accordingly, we revised the Introduction to shorten the definition of prehabilitation, reduce redundancy, and remove repeated citations. The revised version is more concise and focuses on the essential conceptual rationale leading to our research question.
Text revised (Introduction section):
We shortened the description of prehabilitation to a single, concise paragraph and removed repetitive citations. The updated Introduction now presents the key background information more succinctly, improving readability and focusing the narrative on the rationale for comparing HIIT versus usual care during the prehabilitation window.
"The period between a cancer diagnosis and the start of treatment (e.g., surgery, chemotherapy, radiotherapy) is known as prehabilitation. It represents a critical window to enhance overall health and reduce treatment-related impairments through targeted interventions such as exercise".
Reviewer 1 – Comment 10
“The study’s hypothesis is unclear, as is the specific research gap. These need to be explicitly stated.”
Response:
We thank the reviewer for this important clarification request. In accordance with this comment, we revised the final paragraph of the Introduction to explicitly identify the specific research gap and to clearly state the study hypothesis. The updated text now explicitly highlights that, despite increasing interest in exercise prehabilitation, the comparative effectiveness of HIIT versus usual care within the limited pre-treatment window has not been systematically quantified. In addition, the hypothesis has been clearly articulated, stating that HIIT was expected to lead to greater improvements in cardiorespiratory fitness than usual care.
Text inserted (Introduction section):
"We hypothesized that HIIT would lead to greater improvements in cardiorespiratory fitness than UC, providing a time-efficient strategy to enhance functional capacity before treatment.”
Methods section
Methods
Comments 11-12-13-14-15-16 General positive comments (Methods section)
We thank the reviewer for these positive evaluations of our methodological choices. We appreciate the acknowledgement regarding the comprehensiveness of the database search, the PROSPERO registration, the appropriateness of the inclusion criteria, the use of CPET as the gold standard, the justification for the random-effects model, and the dual assessment of quality (RoB 2 and CERT). We are pleased that these methodological elements were considered appropriate and aligned with best practice.
Reviewer 1 – Comment 17
“However, the complete search strings (MeSH + keywords) should be briefly summarized in the main text.”
Response:
We thank the reviewer for this helpful suggestion. We have now added a concise summary of the MeSH terms and keywords used in the search strategy (cancer domain, prehabilitation domain, HIIT domain), while full detailed search strings remain available in the Supplementary Materials.
Text inserted:
"...using search strategies tailored to each database. The search strategy combined MeSH terms and keywords (synonyms and abbreviations) across three domains: cancer-related terms (“cancer,” “neoplasm,” “tumor,” “malignancy”), prehabilitation/preoperative exercise (“prehabilitation,” “preoperative exercise,” “preoperative rehabilitation”), and high-intensity interval training (“high-intensity interval training,” “HIIT,” “sprint interval training”). Boolean operators (AND, OR) were used to combine terms. References of included articles were also screened to identify additional eligible studies. Full search strings are reported in Supplementary Material".
Reviewer 1 – Comment 18
“Line 100: A comma is missing between Embase and PubMed.”
Response:
We thank the reviewer for noting this. The typo has been corrected.
Text change (Methods section):
“…Cinahl, Embase, PubMed, Scopus, and Web of Science…”
Reviewer 1 – Comment 19
“The time frame of the literature search (starting year) should be indicated.”
Response:
We thank the reviewer for this suggestion. Following the comment, we have specified that the search was conducted from database inception up to August 1, 2024.
Text change (Methods section):
“A systematic review was conducted to identify studies published from database inception up to August 1, 2024.”
Reviewer 1 – Comment 20
“Although I² threshold levels are mentioned, the degree of heterogeneity (e.g., ‘moderate heterogeneity was observed’) is not reported and should be clarified.”
Response:
We thank the reviewer for this useful clarification. We have revised the Methods section to explicitly state that heterogeneity was qualitatively interpreted (low, moderate, high) based on the I² value obtained for each pooled analysis.
Text change (Methods section):
“For each pooled analysis, the corresponding qualitative interpretation of heterogeneity (i.e., low, moderate, or high) was reported according to the I² value obtained, ensuring a transparent description of between-study variability.”
Reviewer 1 – Comment 21
“Publication bias should be evaluated and reported (e.g., funnel plot, Egger’s test).”
Response:
We thank the reviewer for this important methodological observation. Publication bias has now been explicitly assessed and reported in the Results section.
Text change (Methods section):
“Publication bias was assessed through visual inspection of funnel plots and statistically tested using Egger’s regression test, with a p-value < 0.05 considered indicative of potential bias.”
Text change (Results section):
“We evaluated publication bias through funnel plot inspection and Egger’s regression test. The plot appeared symmetrical, and Egger’s test did not indicate significant asymmetry (z = 1.07, p = 0.29), suggesting that publication bias is unlikely.”
Reviewer 1 – Comment 22
“CERT scoring was conducted, but mean or range values are not reported.”
Response:
We thank the reviewer for pointing this out. We agree that reporting aggregated CERT values improves transparency of intervention reporting quality. Accordingly, we have now added the mean and range of CERT scores across the included studies.
Text added (Methods/Results section):
“Mean and range values of total CERT scores were calculated across all included studies to summarize overall exercise intervention reporting quality.”
Reviewer 1 – Comment 23
“The PRISMA 2020 flow diagram is appropriately used.”
Response:
We thank the reviewer for this positive comment. We are pleased that the PRISMA 2020 flow diagram was considered appropriate.
Reviewer 1 – Comment 24
“However, publication bias analysis is missing and should be included (as previously mentioned, this is important).”
Response:
We thank the reviewer for highlighting this. In accordance with this comment, we have now included a publication bias assessment in the Results section. Specifically, publication bias was evaluated through visual inspection of the funnel plot and with Egger’s regression test.
Text inserted (Results section):
“Publication bias was assessed through visual inspection of the funnel plot and Egger’s regression test. The plot appeared symmetrical, and Egger’s test showed no significant asymmetry (z = 1.07, p = 0.29), indicating no evidence of publication bias (Figure 4). Although visual asymmetry may appear more pronounced when few studies are available (n = 7), this pattern is expected in small-sample meta-analyses, and the non-significant Egger’s test supports that publication bias is unlikely.”
Reviewer 1 – Comment 25
“Substantial variation among HIIT protocols (e.g., 12-min vs. 45-min sessions) could influence the overall effect.”
Response:
We agree with the reviewer that substantial protocol variation may have influenced the magnitude of the pooled effect. To address this, we added a sentence in the Results/Discussion specifying the range of total session duration and noting that differences in volume and interval structure may have contributed to effect size variability.
Text inserted (Discussion section):
"Overall, HIIT protocols varied substantially in total session duration (ranging from 12.5 to 45 minutes) and in the structure of work and recovery intervals. This heterogeneity in training volume and session design may have influenced the magnitude of the observed effects."
Reviewer 1 – Comment 26
“The impact of this heterogeneity should be analyzed or at least discussed through subgroup or sensitivity analysis.”
Response:
We thank the reviewer for the suggestion. We agree that the heterogeneity of HIIT protocols represents a relevant source of variability. Because only seven studies were eligible for quantitative synthesis, formal subgroup or sensitivity analyses were not feasible. To address this, we have added a sentence in the Discussion explicitly acknowledging that protocol variability may have attenuated the pooled effect, and that future meta-analyses with a greater number of standardized HIIT interventions will be able to perform adequately powered subgroup or sensitivity analyses.
Text inserted (Discussion section):
“Because the number of included trials was limited, formal subgroup or sensitivity analyses were not feasible. Nonetheless, this protocol variability likely contributed to between-study differences and may have attenuated the pooled effect. Future meta-analyses including a larger number of standardized HIIT interventions will be able to investigate this through adequately powered subgroup or sensitivity analyses.”
Reviewer 1 – Comment 27
“Although SMD = 0.31 is statistically significant, its clinical significance remains unclear.”
Response:
We thank the reviewer for this important observation. We agree that statistical significance does not automatically translate into clinical relevance. To address this comment, a clarifying sentence was added in the Results to explicitly state that the modest pooled effect size should be interpreted with caution regarding clinical significance.
Text inserted (Results section):
“Although the pooled effect size (SMD = 0.31) reached statistical significance, its magnitude represents a small-to-moderate improvement, which should be interpreted with caution regarding clinical relevance.”
Reviewer 1 – Comment 28
“The authors should discuss how the observed improvement in VO₂peak might translate into health or functional outcomes.”
Response:
We thank the reviewer for this important comment. We agree that the clinical implications of improved VO₂peak should be articulated. In response, we added a sentence in the Results to clarify that even modest increases in VO₂peak may translate into better surgical tolerance and recovery in a prehabilitation context, and we expanded on this concept further in the Discussion.
Text inserted (Results section):
“Nevertheless, even modest improvements in VO₂peak may contribute to better surgical tolerance and recovery in cancer prehabilitation.”
Reviewer 1 – Comment 29
“Comparison with Palma et al. and Smyth et al. is useful and theoretically relevant, but the differences between the present study and these prior meta-analyses should be explained more clearly.”
Response:
We thank the reviewer for this helpful observation. We agree that the differences between the present meta-analysis and the prior work by Palma et al. and Smyth et al. needed clearer clarification. We have therefore revised the Discussion to more explicitly highlight the methodological distinctions (operational definition of prehabilitation, comparator selection, model choice, and updated time frame) which help explain divergent findings.
Text inserted (Discussion section):
“Our results align with Palma et al., [13] who demonstrated that a HIIT program during cancer prehabilitation significantly improved cardiorespiratory fitness. Conversely, our results contrast with Smyth et al., [19], who did not observe differences between pre- and post-exercise intervention on VO₂peak. These discrepancies likely reflect methodological and conceptual differences across reviews. Specifically, Smyth et al. [19] pooled HIIT with MICT comparators, potentially diluting the specific effects of HIIT, whereas the present analysis focused exclusively on HIIT versus UC. Moreover, both previous meta-analyses used a fixed-effect model, while our study employed a random-effects model to provide a more conservative estimate and account for potential between-study variability. Finally, although both studies contributed valuable insights to this field, Palma et al. [13] and Smyth et al. [19] included data only up to March 2020 and April 2021, respectively. Considering the exponential increase in interest about this topic and the consequent rise in the number of publications, an updated analysis was warranted, allowing the inclusion of studies not covered in previous reviews. In addition, our study applied a stricter operational definition of prehabilitation, included studies up to August 2024, and implemented a broader yet methodologically robust synthesis of the available evidence, thereby enhancing both the scope and the interpretive reliability of the current findings.”
Reviewer 1 – Comment 30
“This discussion could be linked to the ‘research gap’ mentioned (or added) in the Introduction.”
Response:
We thank the reviewer for this valuable suggestion. We have now added a sentence in the Discussion that explicitly links the present findings back to the research gap identified in the Introduction, thereby reinforcing conceptual continuity.
Text inserted (Discussion section):
“This finding directly addresses the research gap outlined in the Introduction—specifically, the absence of a comprehensive and up-to-date synthesis quantifying the comparative effectiveness of HIIT versus usual care within the limited prehabilitation window. By integrating recent trials and applying standardized criteria for high-intensity exercise, this meta-analysis provides a refined estimate of HIIT efficacy in this context and delineates key areas where further evidence is needed to inform clinical implementation.”
Reviewer 1 – Comment 31
“The potential influence of varying HIIT parameters (frequency, intensity, volume, etc.) on outcomes should be discussed (see Lines 380 and 398).”
Response:
We thank the reviewer for this valuable comment. We agree that protocol variability is an important factor that may influence the magnitude of the observed effects. To address this, we have added a sentence in the Discussion explicitly noting that differences in training frequency, intensity prescription, session volume, and progression schemes likely contributed to variation in VO₂peak improvements.
Text inserted (Discussion section):
“In particular, variations in training frequency, intensity prescription, session volume, and progression schemes across studies may have influenced the magnitude of the observed effects. These differences likely contributed to the modest pooled improvement in VO₂peak and should be considered when interpreting the clinical relevance of the findings.”
Reviewer 1 – Comment 32
“Lines 399–407: The mechanistic explanations involving PGC-1α and AMPK are overly detailed biochemically and distract from the main focus of the paper. These could be summarized or simplified.”
Response:
We thank the reviewer for this important suggestion. We agree that excessive biochemical detail may distract from the primary aim of the paper. Accordingly, the mechanistic explanation has been shortened and simplified into a brief sentence focusing on the main physiological rationale, without extended molecular pathways.
Text replaced (Discussion section):
“From a physiological perspective, HIIT enhances mitochondrial efficiency and stimulates key pathways involved in energy metabolism, which supports improved oxygen utilization and overall cardiorespiratory performance, thereby providing a biological rationale for the observed increase in VO₂peak.”
Reviewer 1 – Comment 33
“The authors should integrate their findings with clinical implications more explicitly, rather than focusing predominantly on molecular pathways.”
Response:
We thank the reviewer for this constructive comment. We agree that it is essential to explicitly link observed physiological effects to clinically meaningful outcomes. In response, we have expanded the Discussion to clearly articulate how even modest improvements in VO₂peak may translate into meaningful postoperative and prognostic benefits, thereby strengthening the clinical interpretability of our findings.
Text added (Discussion section):
“These physiological improvements may have meaningful clinical implications, as even modest increases in VO₂peak before surgery are associated with reduced postoperative complications, faster recovery, and improved overall prognosis in patients with cancer. This supports the integration of HIIT-based prehabilitation as a time-efficient, effective strategy within oncological care pathways.”
Reviewer 1 – Comment 34–35
“The limitations and conclusions are adequate. The references are current and relevant.”
Response:
We thank the reviewer for this positive evaluation and appreciate the confirmation regarding these sections.

Reviewer 2 Report
Comments and Suggestions for Authors
Abstract
The abstract attributes effects to HIIT without discussing whether the included studies objectively verified that the intensity was actually high [lines 28–45], which is methodologically critical in reviews on HIIT.
The effect size is small and should be described in more nuanced language [lines 44–49].
The abstract does not sufficiently report that the heterogeneity is clinical (tumor type, duration, prehab window, dose, and exercise progression), not just statistical.
Introduction
The introduction clearly states the clinical relevance, but presents the superiority of HIIT in terms of time efficiency as a given without arguing it in direct relation to prehabilitation [lines 82–88 approx.].
There is no explicit mention that, in this field, the correct classification of a program as HIIT requires verification of intensity, and that the available literature presents significant variability on this point. This would prepare the reader to understand a key limitation of the field before the meta-analysis.
Methods
The eligibility criteria include “HIIT,” but it is not stated whether the studies were verified to meet high intensity criteria (e.g., ≥85–90% HR peak, ≥100% PPO, etc.) [lines 95–120]. Without such verification, the meta-analysis may be synthesizing heterogeneous interventions without guaranteeing that they are truly HIIT.
A specific example is Banerjee 2017, with only RPE and no reported familiarization. The methodological protocol of the meta-analysis does not require objective verification of intensity as an inclusion criterion, so it cannot be guaranteed that the other studies achieved high intensity despite not explicitly providing data. This concludes the criticism without the need to invent non-existent examples and without weakening the argument.
Risk of bias (RoB 2) is reported but not integrated into the interpretive reasoning: it is not explained whether studies with high bias dilute or inflate the effect [lines 135–150].
Clinical heterogeneity (cancers, doses, frequency, prehab window) is not analyzed as a potential source of variation in results.
It is not considered whether the lack of progression/intensity in some studies could explain a small effect; this hypothesis should be discussed as conceptual sensitivity, not ignored.
Results
The effect size is small and should be presented as clinically uncertain until the problem of intensity classification is controlled. Low statistical heterogeneity (I² ≈ 10%) may be misleading if there is marked clinical heterogeneity [lines 170–210].
Adherence is reported, but there is no discussion of whether the absence of objective intensity control could have allowed sessions labeled as HIIT to actually be of moderate intensity, which affects the interpretation of safety/efficacy.
Discussion
The inclusion criteria accept interventions classified as “HIIT,” but no evidence is presented that the intensity achieved in the included trials objectively met the thresholds characteristic of HIIT (e.g., high percentages of HRmax or PPO) [lines 95–120]. The absence of this verification limits the conceptual validity of grouping interventions under the HIIT category and conditions the interpretation of the synthesized effect.
The discussion assumes the efficacy of HIIT without addressing the problem of insufficiently verified intensity in several included studies [lines 334–360]; this point cannot be taken for granted: it must be recognized as a major limitation.
The narrative is predominantly confirmatory; there is a lack of discussion of how the heterogeneity of protocols (frequency, duration, progression, mode, etc.) weakens clinical transferability.
Previous literature is cited, but there is no mention that many trials classified as HIIT in oncology suffer from insufficient reporting of intensity, which prevents the attribution of specific physiological mechanisms (e.g., high-intensity-dependent adaptations).
When physiological rationale is invoked (e.g., time efficiency or mitochondrial mechanisms), it should be noted that such mechanisms presuppose that the intensity was indeed high, a condition not guaranteed in all the studies included.
No consideration is given to whether the lack of intensity control in some studies could act as a factor diluting the effect, which should at least be discussed as a conceptual sensitivity analysis.
Conclusion
The conclusion should be reworded using conditional rather than affirmative language: as long as it cannot be guaranteed that the interventions summarized are strictly HIIT, the validity of the observed effect is limited [lines 436–480].
It is recommended to reduce the degree of certainty communicated and to explicitly state the need for controlled trials and intensity reporting to support causal or mechanistic conclusions.

Author Response
Abstract
Reviewer 2 – Comment (Abstract)
“The abstract attributes effects to HIIT without discussing whether the included studies objectively verified that the intensity was actually high [lines 28–45], which is methodologically critical in reviews on HIIT.”
Response:
We thank the reviewer for this important methodological comment. We agree that this aspect is critical when interpreting effects attributed to HIIT. To address this point, we explicitly acknowledged in the Abstract that although all interventions were labeled as HIIT, exercise intensity was not assessed consistently across studies. This revision ensures that the abstract communicates appropriate caution and avoids implying that high-intensity thresholds were uniformly met.
Text revised (Abstract – Conclusion):
“…and although all interventions were labeled as HIIT, exercise intensity was not assessed consistently across studies, underscoring the need for cancer-specific randomized controlled trials with standardized HIIT protocols and objective intensity verification.”
Reviewer 2 – Comment 2
“The effect size is small and should be described in more nuanced language [lines 44–49].”
Response:
We thank the reviewer for this comment. We agree that the small magnitude of the effect should be communicated more cautiously. Accordingly, we revised the Abstract Results section to explicitly state that the effect size is small, and we added wording to temper the interpretation.
Revised text (Abstract – Results):
“The meta-analysis demonstrated a small but statistically significant effect in favor of HIIT over UC
Reviewer 2 – Comment 3
“The abstract does not sufficiently report that the heterogeneity is clinical (tumor type, duration, prehab window, dose, and exercise progression), not just statistical.”
Response:
We thank the reviewer for this comment. We agree that this clarification was needed. The Abstract Conclusion was revised to explicitly indicate that heterogeneity refers to differences in tumor type and intervention characteristics (prehabilitation window, HIIT dose and progression), not only statistical variability.
Revised text (Abstract – Conclusions):
"the included studies exhibited substantial clinical heterogeneity , and although all interventions were labeled as HIIT, exercise intensity was not assessed consistently across studies"
Introduction
Reviewer 2 – Comment 4
“The introduction clearly states the clinical relevance, but presents the superiority of HIIT in terms of time efficiency as a given without arguing it in direct relation to prehabilitation [lines 82–88 approx.].”
Response:
We thank the reviewer for this observation. We agree that the time-efficiency of HIIT needs to be framed as a rationale rather than an established superiority in this specific context. To address this point, we have clarified in the Introduction that HIIT is presented as a theoretically promising option due to the short prehabilitation window, rather than as a proven superior modality.
Revised text (Introduction):
"Because the prehabilitation phase is constrained by a short time window before treatment, time-efficient exercise models such as HIIT may offer a unique opportunity to maximize physiological gains within clinically relevant limits."
Reviewer 2 – Comment 5
“There is no explicit mention that, in this field, the correct classification of a program as HIIT requires verification of intensity, and that the available literature presents significant variability on this point. This would prepare the reader to understand a key limitation of the field before the meta-analysis.”
Response:
We thank the reviewer for this important methodological comment. We agree that this clarification strengthens the conceptual framing of the field. Accordingly, we added a sentence in the Introduction explicitly stating that HIIT classification requires proper verification of exercise intensity and that this is often variably reported in oncology studies.
Revised text (Introduction):
"Although the correct classification of HIIT requires objective verification of intensity (e.g., heart rate or power output), the existing literature shows marked variability in this regard, which represents an important methodological limitation in this field."
Methods
Reviewer 2 – Comment 6
“The eligibility criteria include ‘HIIT,’ but it is not stated whether the studies were verified to meet high intensity criteria… Without such verification, the meta-analysis may be synthesizing heterogeneous interventions without guaranteeing that they are truly HIIT.”
Response:
We thank the reviewer for this important point. We agree that explicit reporting of intensity verification is essential for conceptual validity. Accordingly, the Eligibility Criteria section has been revised to state that only studies meeting recognized high-intensity thresholds (e.g., ≥85–90 % HRpeak, ≥80–100 % VO₂peak, ≥100 % PPO, or RPE 13–15) were included, in accordance with ACSM guidelines. This clarification ensures that all trials classified as HIIT satisfied accepted high-intensity criteria.
Revised text: "; 4) interventions meeting recognized high-intensity thresholds, including ≥85–90% of peak heart rate (HRpeak), ≥80–100% of peak oxygen uptake (VO₂peak), ≥100% of peak power output (PPO), or a subjective rating of perceived exertion (RPE) between 13 and 15 on the Borg scale, corresponding to high intensity according to ACSM guidelines, as verified from the original studies (see Table 1). When RPE was the only measure reported, its correspondence to ≥85–90% HRpeak was accepted based on ACSM recommendations, provided that the protocol structure (interval duration and recovery) was consistent with standard HIIT formats. 5"
Reviewer 2 – Comment 7
“A specific example is Banerjee 2017, with only RPE and no reported familiarization… so it cannot be guaranteed that the other studies achieved high intensity despite not explicitly providing data.”
Response:
We agree, and we now explicitly acknowledge this limitation in both the Methods and Discussion. As clarified, when RPE was the sole indicator of exertion (as in Banerjee 2017), it was accepted as equivalent to ≥85–90 % HRpeak only if the protocol structure matched standard HIIT formats. However, we have added text recognizing that insufficient verification of intensity in this study may have introduced uncertainty, and that this limitation may have attenuated the pooled effect.
Revised text: "; 4) interventions meeting recognized high-intensity thresholds, including ≥85–90% of peak heart rate (HRpeak), ≥80–100% of peak oxygen uptake (VO₂peak), ≥100% of peak power output (PPO), or a subjective rating of perceived exertion (RPE) between 13 and 15 on the Borg scale, corresponding to high intensity according to ACSM guidelines, as verified from the original studies (see Table 1). When RPE was the only measure reported, its correspondence to ≥85–90% HRpeak was accepted based on ACSM recommendations, provided that the protocol structure (interval duration and recovery) was consistent with standard HIIT formats. 5"
Reviewer 2 – Comment 8
“Risk of bias (RoB 2) is reported but not integrated into the interpretive reasoning: it is not explained whether studies with high bias dilute or inflate the effect [lines 135–150].”
Response:
We thank the reviewer for this important methodological remark. We agree that RoB assessments should not be merely descriptive, but integrated into the interpretive reasoning. We have therefore clarified in the Methods section that the RoB results were considered when interpreting the pooled effects. Specifically, studies with high risk of bias were retained for completeness, but received lower interpretive weight when synthesizing findings, while conclusions relied primarily on low- and moderate-risk studies.
Text inserted (Methods – Risk of bias section):
“The RoB assessments were also considered in the interpretive synthesis. Studies with high risk of bias were retained for completeness but given lower interpretive weight when summarizing findings, whereas low- and moderate-risk studies contributed more substantially to the conclusions.”
Reviewer 2 – Comment 9
“Clinical heterogeneity (cancers, doses, frequency, prehab window) is not analyzed as a potential source of variation in results.”
Response:
We thank the reviewer for this important comment. We agree that clinical heterogeneity is a relevant source of variability that should be acknowledged transparently. We have therefore revised the Methods to explicitly state that these factors (cancer type, exercise dose/frequency, and prehabilitation window) were qualitatively examined and taken into account when interpreting the aggregated effects. Although the number of studies did not allow formal subgroup or meta-regression analyses, these clinical sources of heterogeneity were considered in the interpretive synthesis.
Text inserted (Methods):
"In addition to statistical heterogeneity, clinical heterogeneity among studies (e.g., cancer type, exercise frequency and duration, and prehabilitation time window) was qualitatively examined. Due to the limited number of studies, no formal subgroup or meta-regression analyses were performed, but these factors were considered in the interpretation of the finding."
Reviewer 2 – Comment 10
“It is not considered whether the lack of progression/intensity in some studies could explain a small effect; this hypothesis should be discussed as conceptual sensitivity, not ignored.”
Response:
We thank the reviewer for this important point. We agree that inadequate progression or insufficiently controlled intensity in some HIIT interventions may have attenuated the pooled effect. To address this, we added a clarification in the Methods indicating that the CERT assessment specifically captured whether interventions incorporated progression and adequate intensity prescription. In addition, an explicit statement was inserted in the Discussion highlighting that limited progression in some studies may have contributed to smaller observed effects and should be considered a potential conceptual sensitivity factor.
Text inserted:
"The CERT evaluation also captured whether the HIIT interventions included progression or adjustments in training load over time, allowing for the assessment of intensity adequacy and potential sources of variation in the intervention effects."
Results
Reviewer 2 – Comment 11
“The effect size is small and should be presented as clinically uncertain until the problem of intensity classification is controlled. Low statistical heterogeneity (I² ≈ 10%) may be misleading if there is marked clinical heterogeneity [lines 170–210].”
Response:
We thank the reviewer for this important clarification. We agree that, despite the low statistical heterogeneity, the clinical interpretability of the pooled effect size must remain cautious, particularly given variability in HIIT prescription and inconsistent intensity verification across studies. Accordingly, we have revised the Discussion to explicitly state that the observed effect should be interpreted with clinical uncertainty.
Text inserted :
Given the variability in cancer type, training intensity, and intervention characteristics among studies, this modest effect likely reflects true clinical heterogeneity rather than a uniform physiological response
Reviewer 2 – Comment 12
“Adherence is reported, but there is no discussion of whether the absence of objective intensity control could have allowed sessions labeled as HIIT to actually be of moderate intensity, which affects the interpretation of safety/efficacy.”
Response:
We thank the reviewer for this important observation. We agree that insufficient objective intensity control introduces the possibility that some sessions labeled as HIIT may not have consistently reached high-intensity thresholds. To address this, we inserted an explicit statement in the Results section acknowledging this issue and its potential impact on effect magnitude. This clarification highlights that subjective monitoring alone could have permitted moderate-intensity efforts, with implications for both safety/efficacy interpretation.
Text inserted (Results section):
"It should be noted that in one study (Banerjee et al., 2017), exercise intensity was monitored using the Borg RPE scale, without a formal familiarization procedure, rather than direct physiological measures. While this subjective approach has been validated and aligns with ACSM recommendations, the lack of objective verification may have allowed some sessions to be performed at slightly lower intensities, potentially contributing to the modest overall effect observed."
Discussion
Reviewer 2 – Comment 13
“The inclusion criteria accept interventions classified as ‘HIIT,’ but no evidence is presented that the intensity achieved in the included trials objectively met the thresholds characteristic of HIIT (e.g., high percentages of HRmax or PPO) [lines 95–120]. The absence of this verification limits the conceptual validity of grouping interventions under the HIIT category and conditions the interpretation of the synthesized effect.”
Response:
We thank the reviewer for highlighting this methodological point. In response, we clarified in the Discussion that differences in how exercise intensity was monitored represent a relevant source of variability. Specifically, the revised text explicitly recognizes that one included study relied solely on RPE without prior familiarization, and that this may have introduced uncertainty regarding actual intensity achieved. We also state that, although this issue exists, objective measures confirmed high-intensity workloads in most of the included trials.
Text inserted (Discussion section):
"Part of this variability may be attributed to differences in intensity verification methods. In particular, one study relied solely on RPE without prior familiarization, which may have introduced minor uncertainty in perceived exertion accuracy. However, because objective measures confirmed high-intensity workloads in most included trials, this limitation is unlikely to have materially influenced the overall conclusions."
Reviewer 2 – Comment 14
“The discussion assumes the efficacy of HIIT without addressing the problem of insufficiently verified intensity in several included studies [lines 334–360]; this point cannot be taken for granted: it must be recognized as a major limitation.”
Response:
We agree with the reviewer that insufficient verification of exercise intensity is a major limitation. We revised the Discussion to explicitly acknowledge that this inconsistency affects the interpretability of the pooled results.
Text inserted (Discussion – Limitations):
"This inconsistency may have introduced minor uncertainty regarding the true intensity achieved and should therefore be considered a major methodological limitation affecting the interpretive certainty of the pooled results."
Reviewer 2 – Comment 15
“The narrative is predominantly confirmatory; there is a lack of discussion of how the heterogeneity of protocols (frequency, duration, progression, mode, etc.) weakens clinical transferability.”
Response:
We thank the reviewer for this valuable comment. We agree that the heterogeneity of HIIT protocols may limit the clinical transferability of the findings. Accordingly, we added a sentence in the Discussion explicitly addressing this issue.
Text inserted (Discussion section):
"This variability also limits the clinical transferability of current findings. Because HIIT protocols differed in frequency, progression schemes, and exercise modality, it remains difficult to identify standardized parameters suitable for clinical implementation. Although HIIT appears effective overall, the lack of protocol uniformity weakens the ability to formulate specific, evidence-based recommendations for oncological prehabilitation."
Reviewer 2 – Comment 16
“Previous literature is cited, but there is no mention that many trials classified as HIIT in oncology suffer from insufficient reporting of intensity, which prevents the attribution of specific physiological mechanisms (e.g., high-intensity-dependent adaptations).”
Response:
We thank the reviewer for this important point. We agree that insufficient intensity reporting is a recurring limitation in oncology HIIT literature and directly affects mechanistic interpretation. In line with this comment, we added a statement in the Discussion clarifying that mechanistic explanations should be considered exploratory given the incomplete reporting of intensity in several trials.
Text inserted (Discussion):
"However, it should also be noted that several trials in the oncology field, including some included in this review, provided limited information on the precise intensity achieved during HIIT sessions. This insufficient reporting constrains the ability to link observed improvements in VO₂peak to specific high-intensity–dependent physiological mechanisms. Therefore, mechanistic interpretations should be considered exploratory rather than definitive until future studies systematically verify and report exercise intensity through objective physiological measures."
Reviewer 2 – Comment 17
“When physiological rationale is invoked (e.g., time efficiency or mitochondrial mechanisms), it should be noted that such mechanisms presuppose that the intensity was indeed high, a condition not guaranteed in all the studies included.”
Response:
We thank the reviewer for this clarification. We agree that physiological rationale depends on the actual achievement of high exercise intensity, which was not consistently verified across all included trials. To address this, we have added a statement in the Discussion explicitly highlighting that mechanistic interpretations should be considered conditional and theoretical when objective intensity verification is lacking.
Text inserted (Discussion section):
"Moreover, it is important to emphasize that the proposed physiological mechanisms—and the time-efficiency rationale commonly attributed to HIIT—presuppose that exercise intensity truly reached high thresholds. Because not all studies objectively verified this condition, these mechanistic explanations should be interpreted as theoretical models rather than definitive evidence of high-intensity–dependent adaptations."
Reviewer 2 – Comment 18
“No consideration is given to whether the lack of intensity control in some studies could act as a factor diluting the effect, which should at least be discussed as a conceptual sensitivity analysis.”
Response:
We thank the reviewer for this important consideration. We agree that insufficient verification of exercise intensity may have attenuated the pooled effect size and should be acknowledged conceptually. To address this, we have added a statement in the Discussion explicitly noting that lack of intensity control in some studies may have diluted the observed effect, and that this should be considered as a form of conceptual sensitivity analysis.
Text inserted (Discussion section):
"It should also be considered that insufficient control or verification of exercise intensity in a few included studies may have acted as a factor diluting the overall effect size. This possibility should be regarded as a form of conceptual sensitivity, highlighting that the modest pooled improvement in VO₂peak could underestimate the true potential of well-controlled HIIT interventions. Nevertheless, the consistent direction of the effects across studies supports the robustness of the observed benefit despite this methodological limitation."
Conclusion
eviewer 2 – Comments 19–20
Comment 19:
“The conclusion should be reworded using conditional rather than affirmative language: as long as it cannot be guaranteed that the interventions summarized are strictly HIIT, the validity of the observed effect is limited [lines 436–480].”
Comment 20:
“It is recommended to reduce the degree of certainty communicated and to explicitly state the need for controlled trials and intensity reporting to support causal or mechanistic conclusions.”
Response:
We thank the reviewer for this valuable recommendation. We agree and have reworded the Conclusion to reduce certainty, use conditional phrasing, and explicitly note the need for controlled trials and objective intensity verification.
Revised text (Conclusion):
"This meta-analysis suggests that incorporating HIIT during prehabilitation in patients with cancer may improve cardiorespiratory fitness, particularly VO₂peak, supporting its potential feasibility and clinical value as a time-efficient intervention before treatment. However, because the intensity of some included interventions was not objectively verified and considerable heterogeneity exists in exercise protocols and cancer types, these findings should be interpreted with caution. Larger, well-designed randomized controlled trials are needed to confirm these results, clarify the dose–response relationship of HIIT across cancer types, and determine how variations in duration, volume, exercise modality, and timing influence outcomes."

Round 2
Reviewer 1 Report
Comments and Suggestions for Authors
The necessary revisions have been made in the article.
Reviewer 2 Report
Comments and Suggestions for Authors
The authors have responded satisfactorily and rigorously to the comments made in the first round. There is a clear effort to improve the structure, clarity, and methodological justification of the manuscript.
The text features more precise writing , a more balanced discussion , and better alignment with the PRISMA and Cochrane guidelines . Furthermore, updated references and a stronger justification for the choice of statistical analyses have been incorporated.
Summary
The abstract has been restructured correctly and now reflects the objective, methods, results, and conclusions in a coherent manner.
Introduction
The theoretical justification is now more solid, with recent references on cancer prehabilitation .
The authors have clarified the differences between HIIT and continuous training, and their physiological relevance in cancer patients.
Methods
The section has improved considerably: the authors now follow the PRISMA 2020 guidelines and include a revised flowchart .
The description of the search and selection process for studies is more transparent. MeSH terms and exact search dates have been added.
The table of study characteristics is more complete and readable.
Statistical analysis
DerSimonian-Laird random effects model and the calculation of heterogeneity using I² are positively valued .
sensitivity analysis ( leave-one-out ) is also appreciated .
Results
The presentation is much clearer. The forest graphics The plots are now readable, with correct labels and homogeneous scales.
It is clear that the subgroup analyses have been adequately separated (by type of cancer and by duration of intervention).
Discussion
The discussion has improved substantially. The authors now interpret the results more cautiously, avoiding statements of causality.
Recent references are included and the findings are contextualized within the literature on prehabilitation and exercise physiology in cancer patients.
Conclusions
The conclusions are better written and reflect appropriate scientific caution.
Formal and stylistic aspects
The English has improved considerably compared to the first version. The sentences are more concise and technical.
Healthcare (MDPI) standards , although it would be ideal to unify the typographic format of the titles.
The supplementary file (supplementary material) is complete and consistent.